# A cell-autonomous tumour suppressor role of RAF1 in hepatocarcinogenesis

Ines Jeric[1], Gabriele Maurer[1,*,†], Anna Lina Cavallo[1,*,†], Josipa Raguz[1], Enrico Desideri[1], Bartosz Tarkowski[1,†], Matthias Parrini[1,†], Irmgard Fischer[1], Kurt Zatloukal[2] & Manuela Baccarini[1]

Hepatocellular carcinoma (HCC) is a leading cause of cancer deaths, but its molecular heterogeneity hampers the design of targeted therapies. Currently, the only therapeutic option for advanced HCC is Sorafenib, an inhibitor whose targets include RAF. Unexpectedly, RAF1 expression is reduced in human HCC samples. Modelling RAF1 downregulation by RNAi increases the proliferation of human HCC lines in xenografts and in culture; furthermore, RAF1 ablation promotes chemical hepatocarcinogenesis and the proliferation of cultured (pre)malignant mouse hepatocytes. The phenotypes depend on increased YAP1 expression and STAT3 activation, observed in cultured RAF1-deficient cells, in HCC xenografts, and in autochthonous liver tumours. Thus RAF1, although essential for the development of skin and lung tumours, is a negative regulator of hepatocarcinogenesis. This unexpected finding highlights the contribution of the cellular/tissue environment in determining the function of a protein, and underscores the importance of understanding the molecular context of a disease to inform therapy design.

[1] Department of Microbiology, Immune biology, and Genetics, Center for Molecular Biology, University of Vienna—Max F. Perutz Laboratories, Doktor-Bohr-Gasse 9, Vienna 1030, Austria. [2] Institute of Pathology, Medical University of Graz, Graz A-8036, Austria. * These authors contributed equally to this work. † Present addresses: AOP Orphan Pharmaceuticals, 1160 Vienna, Austria (G.M.); Institute of Animal Breeding & Genetics, University of Veterinary Medicine, 1210 Vienna, Austria (M.P.); AstraZeneca R&D Discovery Sciences RAD, SE-431 83 Mölndal, Sweden (A.L.C.); Laboratory of Molecular and Cell Neurobiology, International Institute of Molecular and Cell Biology, 02-109 Warsaw, Poland (B.T.). Correspondence and requests for materials should be addressed to M.B. (e-mail: manuela.baccarini@univie.ac.at).

Hepatocellular carcinoma (HCC) is the commonest primary liver malignancy and the fifth most frequent cancer death cause in men. The incidence is highest in developing countries but cases in the Western World are increasing. The 5-year survival rate is poor, biomarkers and molecule-based therapies are lacking, and resistance to currently used chemotherapies is common. HCC correlates with hepatitis virus B or C infection, but also with exposure to aflatoxin B, alcohol abuse and obesity[1]. Liver injury is a strong proliferative stimulus for surviving hepatocytes, which re-enter cell cycle to maintain organ mass and function. Injury/regeneration cycles favour the accumulation of genetic alterations and thus oncogenic hepatocyte transformation, ultimately leading to liver cancer. Activation of the Wnt/βcatenin pathway combined with oxidative stress metabolism and RAS/ERK pathway, loss of tumour suppressor genes, and mutations in chromatin regulators are most frequently observed; overexpression or activation of receptor tyrosine kinases such as ERB2 and MET, of the mTOR pathway, as well as CMYC (ref. 2) and the transcriptional co-activator YAP1 (ref. 3), are observed with varying frequency. Activating mutations of the interleukin 6 (IL6) receptor subunit GP130 and of the transcription factor STAT3 are frequent in inflammatory HCC[4,5].

Hepatocarcinogenesis can be recapitulated in the mouse, allowing functional analysis of specific signalling pathways. Genetic manipulation of JNK and p38 MAPK or the NF-kB pathway induce hepatocarcinogenesis or accelerate chemically driven tumorigenesis by increasing hepatocyte apoptosis, compensatory proliferation and/or inflammation[6]; pathways converging on STAT3 promote the progression of premalignant cancer progenitor cells[7]. Finally, the Hippo pathway and its target YAP1 are key regulators of hepatocyte differentiation in tumourigenesis[3].

RAF1 is a kinase best known as the effector linking RAS to MEK/ERK activation. Additional essential functions of RAF1 rely on protein–protein interaction-based cross-talk with other pathways including Hippo, whose function is antagonized by RAF1 (ref. 8). In the mouse, Raf1 ablation causes liver apoptosis[9,10], suggesting an essential function in this organ and a potential role in liver cancer development. Contrary to this expectation, patient data show reduced RAF1 expression in human HCCs; based on this, we have investigated the role of RAF1 in HCC using two different mouse models: (1) HCC xenografts and (2) hepatocarcinogenesis induced by the alkylating agent diethylnitrosamine (DEN) and promoted by Phenobarbital (Pb), which mimics human disease in terms of gene expression profiles and critically depends on inflammation[11–14]. Both models have revealed a tumour suppressor function of RAF1 in HCC, consistent with the reduced RAF1 expression in HCC patients.

## Results

**Loss of RAF1 promotes HCC development**. We analysed RAF1 expression in paired tumour and non-tumour tissue of each of 31 human HCC specimens. RAF1 expression in tumours was significantly lower compared with the matched surrounding non-tumour tissue, and the degree of RAF1 expression in tumour (defined as the ratio of RAF1 expression in matched tumour/non-tumour tissues) negatively correlated with tumour grade (Fig. 1a). This was surprising for us but it is backed up by the data in the protein atlas, showing that RAF1 expression is low or undetectable in HCC samples probed with two different antibodies (http://www.proteinatlas.org/ENSG00000132155-RAF1/cancer/tissue/liver + cancer).

To determine whether RAF1 ablation plays a role in HCCs as suggested by the analysis of these data, we generated an isogenic

HCC cell line in which RAF1 can be knocked down by the expression of a shRNA controlled by a doxycycline-inducible promoter, without affecting the expression of A- or BRAF (Hep3B RAF1 KD; Fig. 1b). Addition of doxycycline to the medium strongly increased the proliferation of Hep3B RAF1 KD cells (Fig. 1b). More importantly, a tremendous increase in tumour mass was observed in HCC xenografts in nude mice when RAF1 was knocked down in vivo by adding doxycycline to the drinking water (Fig. 1c).

Finally, we examined the role of RAF1 in the development of autochthonous liver tumours using conditional RAF1 ablation in hepatocytes (parenchymal cells) and bile duct cells (AlfpCre;Raf1[F/F] mice, heretofore referred to as Δhep) as well as global deletion of RAF1 by injecting MxCre;Raf1[F/F] mice with Poly I:C (ref. 15) (termed Δp/np; RAF1 ablated in parenchymal and non-parenchymal liver cells). Deletion was efficient and did not affect the expression of other RAF kinases (Supplementary Fig. 1a and b). The mice did not develop spontaneous liver tumours but were more sensitive than controls to DEN/Pb-induced hepatocarcinogenesis. A significant increase in macroscopic tumours was obvious in Δhep livers 30 weeks after DEN treatment; liver:body weight ratio, tumour numbers and tumour-occupied area were significantly higher in Δhep mice than in controls (Fig. 1d). Most lesions were adenomas comprising hepatocytes with a high nuclear:cytoplasmic ratio and compressing the surrounding liver parenchyma; however, 4 out 6 Δhep animals developed HCC (showing trabecular growth patterns, high cellularity and low nuclear:cytoplasmic ratio), compared with 1 out of 6 controls. Consequently, the survival rate of Δhep mice was decreased (Supplementary Fig. 1c). In Δp/np mice, liver:body weight ratios and tumour-occupied areas were only slightly increased; however, RAF1 ablation increased tumour multiplicity (Fig. 1e) and malignancy (7 out of 10 Δp/np animals developed HCC, compared with 3 out of 8 F/F mice). Thus, RAF1 suppresses chemical hepatocarcinogenesis.

Analysis of both models revealed low, similar numbers of apoptotic cells in control and Δhep or Δp/np tumour-bearing livers (Supplementary Fig. 1d). Mitotic indexes were similar in tumours of all genotypes (Fig. 1e). However, more cycling cells, mostly non-parenchymal (Np cells), were present in the non-tumour tissue of Δhep, but not Δp/np livers (Fig. 1f, Supplementary Fig. 1f). The dominant non-parenchymal cell type in livers of all genotypes were F4/80[+] cells, mostly concentrated around portal veins; Δhep livers contained more macrophages, granulocytes and CD3[+] cells than controls (Fig. 1g), while this increase was not observed in Δp/np livers (Supplementary Fig. 1h). Numbers of F4/80[+] cells were similar in tumours of all genotypes (Supplementary Fig. 1g). Thus, RAF1 ablation in parenchymal cells led to increased inflammation in tumour-bearing livers. Consistent with this, the monocyte chemoattractant CCL2 was elevated in the blood of Δhep animals (Fig. 1h), and Δhep livers contained increased amounts of chemokines (CCL2, 4, 5 and 7; CXCL1) and cytokines (IL2, IL4, IL5, IL6, IL10, IL27, TNFα, IL1β, IFNγ; Fig. 1i). In contrast, the levels of both serum chemokines and liver chemokines/cytokines detected in Δp/np mice were comparable to those of controls, except CCL2 which was slightly elevated (Supplementary Fig. 1i and j). As in most chemical models of hepatocarcinogenesis[16], fibrosis or cirrhosis were not detected (Supplementary Fig. 1k). Thus, RAF1 ablation in hepatocytes increased tumour multiplicity, whereas lack of RAF1 in non-parenchymal cells restrained inflammation leading to reduced tumour size.

Carcinogenesis in the DEN model critically depends on the interplay between hepatocytes and inflammatory macrophages[13]. These cells express FSP1, a protein involved in macrophage recruitment and chemotaxis in vivo[17], as well as high amounts of

chemokines and cytokines[18]. FSP1[+] cells increased during carcinogenesis in livers of all genotypes; however, the increase in Δhep organs was significantly higher than in control and Δp/np livers (Supplementary Fig. 2a and b). Consistent with the slightly elevated CCL2 levels in tumour-bearing Δp/np mice, RAF1-deficient hepatocytes expressed higher basal and LPS-induced levels of this chemokine (Supplementary Table 1). However, RAF1-deficient macrophages failed to migrate into matrigel plugs containing CCL2 in vivo (Supplementary Fig. 2c) or in a transwell assay in culture (Supplementary Fig. 2d); treatment with low concentrations of a chemical inhibitor of the RAF1 interaction partner ROKα (ref. 8) restored migration

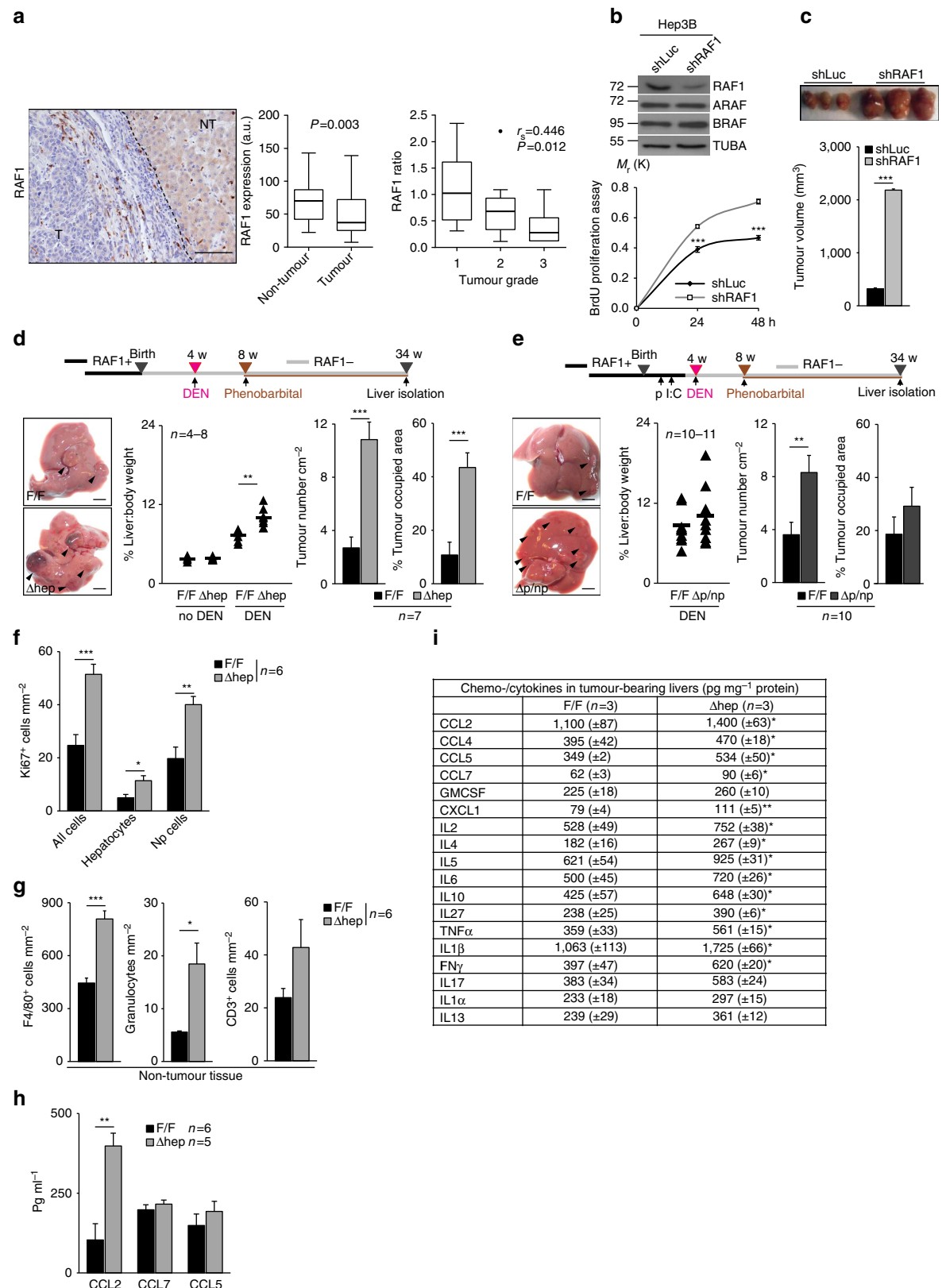

(Supplementary Fig. 2d and e). RAF1-deficient and proficient macrophages produced similar amounts of chemo- and cytokines (Supplementary Table 1). These data imply that widespread RAF1 ablation impairs the recruitment of macrophages to the tumour-bearing Δp/np livers, limiting the inflammatory reaction and restraining the growth, but not the onset of DEN/Pb-induced tumours originating from RAF1-deficient hepatocytes.

**RAF1 ablation increases the number of cancer progenitor cells**. We next determined the impact of RAF1 on the early stages of chemical hepatocarcinogenesis. Liver damage and the number of apoptotic and cycling cells following DEN administration were indistinguishable in Δhep, Δp/np and F/F organs (Supplementary Fig. 3a–c). Later during tumourigenesis, however, more cycling hepatocytes could be detected in RAF1-deficient livers (8 weeks after carcinogen application in Δhep and 12 weeks in Δp/np livers; Fig. 2a). In addition, foci of altered hepatocytes[19] (http://www.niehs.nih.gov/research/resources/liverpath/ hyperplast/index.cfm#focicellular) were observed in all genotypes. These foci contained Ki67 + cells and cells expressing the liver cancer progenitor marker CD44 (ref. 7), and increased YAP1, a key negative regulator of hepatocyte differentiation[20] (Fig. 2b). The foci were more frequently observed in RAF1-deficient than in F/F livers. Thus, conditional ablation of RAF1 by two different Cre transgenes accelerates hepatocarcinogenesis. The faster proliferative reaction of Δhep mice is likely due to increased inflammation (exemplified by macrophage recruitment, Supplementary Fig. 2a). To investigate the role of hepatocyte RAF1 in an inflammatory environment similar to that of F/F mice (Supplementary Fig. 2b, Supplementary Fig. 1j), we concentrated on the Δp/np model and quantified the number of liver cancer progenitor cells 16 weeks after DEN treatment. Progenitors express high levels of the adhesion molecule CD44, which promotes cell–cell adhesion, and are therefore mostly found in aggregate-containing liver fractions[7]. The number of liver cancer progenitor cells (defined as CD44$^+$/CD31$^-$ Ter119$^-$ CD45$^-$) was significantly increased in RAF1-deficient organs (Fig. 2c).

**Molecular characterization of RAF1-deficient lesions**. We next analysed the state of signalling pathways connected to RAF1 and HCC development in tumour-bearing livers and in xenografts. In F/F tumour-bearing livers, ERK phosphorylation was observed in non-tumour, but not in tumour tissue; in Δp/np livers, ERK phosphorylation could also be detected in tumours, implying that RAF1 is dispensable for ERK activation under these conditions (Fig. 3a). RAF1 interacts with the mammalian Hippo pathway[21], a prominent suppressor of hepatocarcinogenesis and of YAP1 activity[3]. Expression of the RAF1 binding partner MST2 and

phosphorylation of MST1/2 were comparable in both genotypes; LATS1 was expressed at slightly higher levels in tumours of both genotypes, and no correlation could be established between RAF1 expression and the phosphorylation of LATS1 on T1079 in the hydrophobic motif. Consistently, although YAP1 expression was higher in tumours and highest in the Δp/np organs, phosphorylation on the Hippo target site S127 was not significantly altered (Fig. 3a). STAT3 Y705 phosphorylation was consistently higher in Δp/np than in F/F tumours and correlated with the expression of the GP130 subunit of the IL6 receptor (Fig. 3a), which has been implicated in gastrointestinal tumourigenesis[22]. The expression of βcatenin, frequently activated in HCC, was slightly elevated in tumours of both genotypes (Fig. 3a); in addition, its subcellular localization was similar in RAF1 deficient and proficient tumours (Supplementary Fig. 4a). Increased YAP1 and GP130 expression, STAT3 and ERK phosphorylation were also observed in Δhep tumour-bearing livers (Supplementary Fig. 4b). In this setting, increased STAT3 phosphorylation could also be observed in the Δhep non-tumour tissue, likely a result of the inflammatory reaction in these organs.

The biochemical phenotypes observed in the autochthonous tumours could be secondary, that is, RAF1 ablation could promote the development of tumours with these molecular characteristics. To investigate this, we performed biochemical analysis of xenograft lysates, which confirmed RAF1 knockdown and recapitulated the increased expression of YAP1 and GP130 as well as the higher levels of STAT3 phosphorylation observed in the autochthonous tumours. ERK phosphorylation, however, was reduced in this setting, suggesting that the increased growth of RAF1 xenografts is ERK independent (Fig. 3b).

We next interrogated the non-selected HCC patient cohort for YAP1 expression. In contrast to RAF1 (Fig. 1a), YAP1 protein levels were higher in tumours compared with the surrounding tissue, and the degree of YAP1 expression in tumours (ratio of YAP1 expression in matched tumour/non-tumour tissue) positively correlated with tumour grade (Fig. 3c). In addition, the ratio of RAF1/YAP1 expression in the same tumour negatively correlated with histological grade in the whole cohort (Fig. 3c), indicating that tumours with low RAF1, high YAP1 protein levels are found in the most malignant group. This is consistent with the higher percentage of HCC-like tumours observed in the chemical carcinogenesis models (see above). We also determined STAT3 expression and nuclear localization (as a proxy for phosphorylation, to avoid possible misrepresentation due to the different fixation/storage conditions[23,24]) in the archival sections at our disposal. STAT3 expression was significantly lower in tumours than in non-tumour tissue (Fig. 3d); this was surprising for us but it is again consistent with the data in the protein atlas,

**Figure 1 | RAF1 is expressed at low levels in human HCC and suppresses the growth of both HCC xenografts and chemically induced tumours.** (**a**) RAF1 expression in a cohort of 31 HCC patients. Left panel, representative IHC image (T, tumour; NT, non-tumour). Scale bar, 50 μm. Middle panel, RAF1 expression in matched tumour and non-tumour tissue (a.u. = arbitrary units). Right panel, RAF1 expression in tumours correlates inversely with tumour grade (ratio: protein expression in tumour/non-tumour tissue). (**b**) Inducible shRNA-mediated RAF1 silencing does not impact A- or BRAF expression (top panel) but increases the proliferation of Hep3B cells in culture (bottom panel; $n = 6$). (**c**) Inducible shRNA-mediated RAF1 silencing strongly promotes the growth of Hep3B xenografts. (**d,e**) Ablation of RAF1 in liver parenchymal cells promotes chemically induced hepatocarcinogenesis. Top panels, experimental protocols. Left bottom panel, macroscopic appearance of F/F and Δhep (**d**) or Δp/np (**e**) tumour-bearing livers 30 weeks (w) after DEN injection; arrows indicate tumours. Scale bars, 0.5 cm. Middle panels, liver:body weight ratio of untreated or DEN/Pb-treated mice. Right panels, tumour numbers and % of tumour-occupied area in control, Δhep (**d**) and Δp/np (**e**) livers. In (**d**), no DEN: F/F $n = 4$, Δhep = 6; DEN-treated: F/F $n = 7$, Δhep = 8. In (**e**), DEN-treated: F/F $n = 10$, Δp/np $n = 11$. (**f,g**) Quantification of Ki67 + cells: (**f**) and inflammatory cells (**g**; F4/80 + cells, granulocytes and CD3 + cells) in tumour-bearing F/F and Δhep livers. Np = non-parenchymal cells. (**h**) Chemokine levels in the serum of F/F and Δhep mice. (**i**) Chemo-/cytokine levels in tumour-bearing livers. Data are presented as mean ± SEM, *$P \le 0.05$, **$P < 0.01$, ***$P < 0.005$. In the box and whiskers plots (Tukey method), the box represents interquartile range, the middle bar the median, and the whiskers extend to 1.5 times the interquartile range. In (**a**), non-tumour versus tumour comparisons were analysed using paired Wilcoxon signed-rank test. $P$ values are indicated in the graph (middle panel); in the right panel data were analysed using Spearman correlation. Spearman's rank correlation coefficient ($r_s$) and $P$ values are indicated within the graph. See also Supplementary Figs 1 and 2 and Supplementary Tables 1 and 2.

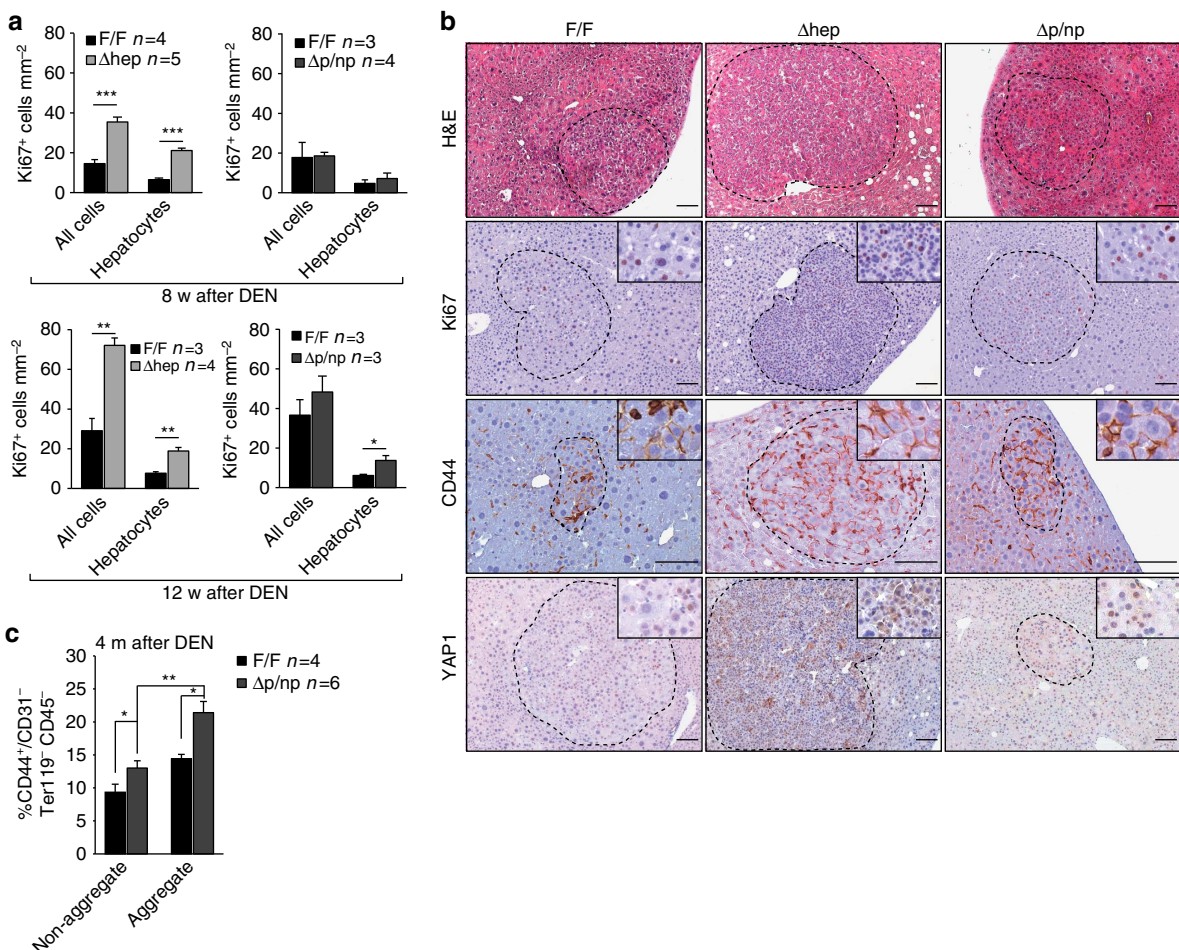

**Figure 2 | RAF1 ablation increases the number of cancer progenitor cells.** (**a**) Quantification of Ki67 + liver cells 8 (top panel) or 12 weeks (bottom panel) after DEN treatment. (**b**) Foci of altered hepatocytes (FAH) in F/F and Δhep or Δp/np livers isolated 12 weeks after DEN injection. Sections were stained with H&E or with the indicated antibodies. FAH are delimited by dotted circles ($n = 3$ per genotype). Scale bars, 50 μm. (**c**) Percentage of cancer progenitor cells (CD44 + /CD31 − Ter119 − CD45 − ) present in non-aggregate and aggregate fractions of F/F and Δp/np livers, as determined by FACS analysis. Data are represented as mean ± s.e.m., *$P \leq 0.05$, **$P < 0.01$, ***$P < 0.005$ according to Student's $t$ test. See also Supplementary Fig. 3.

showing that STAT3 expression is weak in cancer in general and undetectable in 40–83% of HCC samples with 4 out of 5 antibodies used (http://www.proteinatlas.org/ENSG00000168610-STAT3/cancer/tissue/liver + cancer). As previously shown for pSTAT3 (ref. 25), STAT3 nuclear staining was restricted to few cells in most of the samples; however, the ratio of RAF1/YAP1 expression in the same tumour negatively correlated with the presence of medium to large clusters of STAT3 nuclear staining. Thus, tumours with low RAF1, high YAP1 expression contain larger clusters of nuclear STAT3.

**RAF1 knockout cells have a competitive proliferation advantage.** Hep3B cells in which RAF1 was silenced by shRNA proliferated better than control cells (Fig. 1b) and showed increased YAP1 and GP130 expression as well as STAT3 activation when grown as xenografts. Similar results were obtained by silencing RAF1 in cultured Hep3B, HuH-7 and HepG2 cells with siRNAs targeting regions distinct from the one targeted by the shRNA (Fig. 4a). Thus, RAF1 silencing impacts the proliferation and signalling in transformed liver cells *in vivo* (Fig. 1c) and in culture (Figs 1b and 4a). To test whether RAF1 ablation confers a competitive proliferation advantage to premalignant liver cells, we established DEN-induced hepatocyte (DIH) lines from *Raf1*^F/F and *MxCre;Raf1*^F/F mice. Following immortalization, both lines were

treated with IFNβ, leading to RAF1 deletion in the *MxCre;Raf1*^F/F DIH (Δ/Δ DIH). As already observed in livers, RAF1 deletion was efficient and did not affect the expression of A- or BRAF (Fig. 4b). RAF1-deficient and control DIH were much smaller than primary hepatocytes (P-HEPS) and expressed CD44 and α-fetoprotein (AFP) instead of albumin (ALB) (Fig. 4c and d). Δ/Δ DIH had a clear proliferation advantage, particularly in media with low serum (Fig. 4e). They also attracted macrophages and produced CCL2 and CXCL1 more efficiently than F/F DIH (Supplementary Fig. 5a and b).

To gain insight into the molecular mechanisms underlying the proliferation phenotype of RAF1-deficient cells, we treated F/F and Δp/np P-HEPS and F/F or Δ/Δ DIH with IL6, a cytokine pivotal for hepatocarcinogenesis[6]. RAF1-deficient cells showed decreased ERK phosphorylation (slight decrease in P-HEPS), but constitutively higher YAP1 and GP130 expression and increased STAT3 phosphorylation (Fig. 4f). Similar results were also observed by knocking down RAF1 in F/F DIH by siRNA (F/F DIH siRAF1, Supplementary Fig. 5c–f), ruling out possible artifacts due to the separate immortalization of F/F and Δ/Δ DIH. YAP1 and GP130 expression was not increased in RAF1-deficient keratinocytes, mouse embryonic fibroblasts, endothelial cells or macrophages (Supplementary Fig. 6), indicating that the mechanism underlying the RAF1-dependent regulation of these proteins is selectively active in hepatocytes.

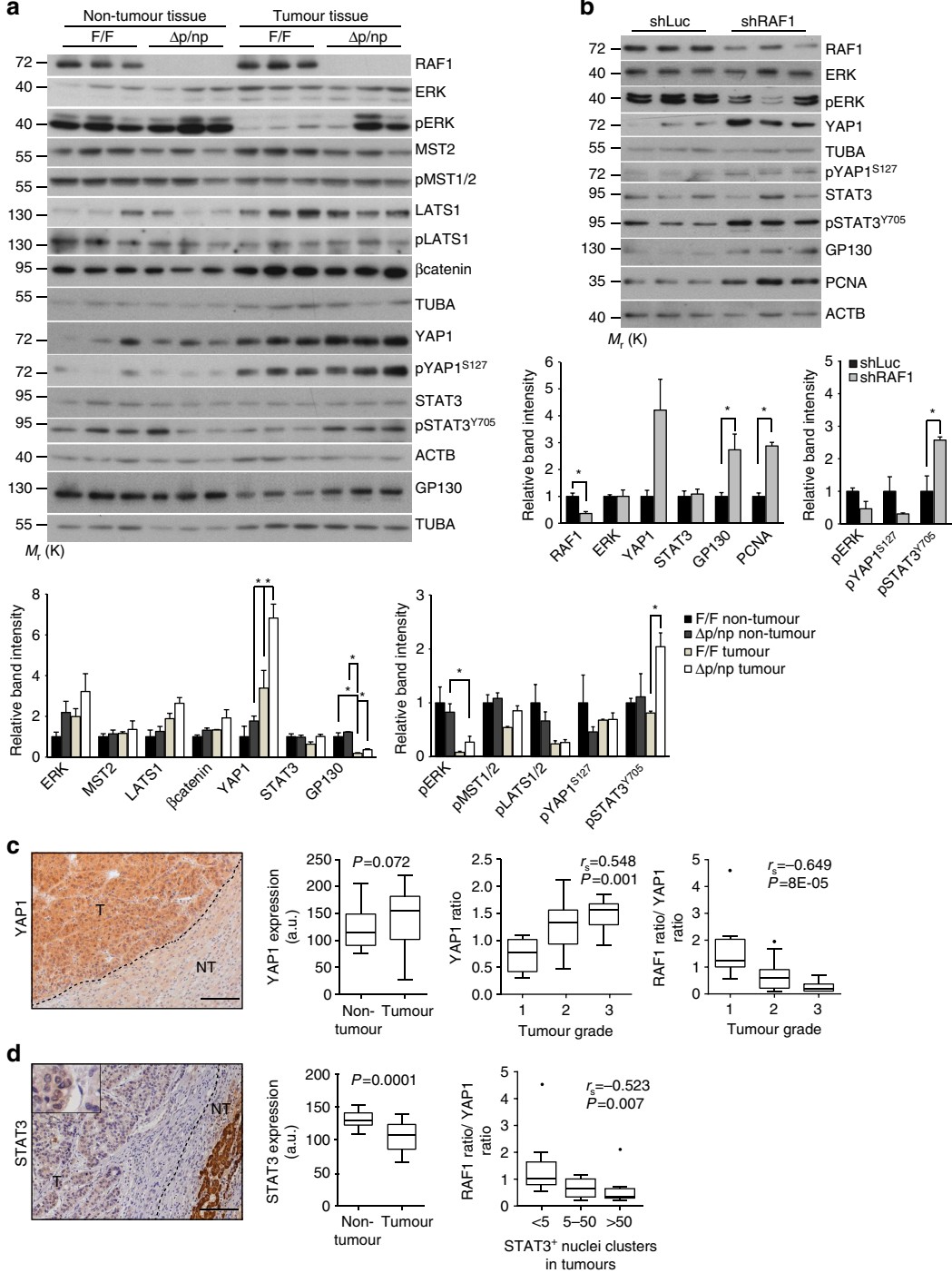

**Figure 3 | Molecular characterization of RAF1-deficient lesions. (a)** Immunoblotting of F/F and Δp/np livers collected 30 weeks after DEN treatment. The plots represents a densitometric quantification of the immunoblot performed using ImageJ. The data are expressed as relative band intensity adjusted to TUBA or ACTB, which serve as loading controls (upper plot). Phosphorylation is expressed as the ratio between the phosphospecific antibody signal and the signal obtained with the protein-specific antibody. In both cases, the data are normalized to the F/F non-tumour samples, which were arbitrarily set as 1. **(b)** Immunoblot analysis of signaling pathways in xenograft samples ($n = 3$, analysed 40 days after transplant). The plots show a quantification of the immunoblots performed as described in (**a**). **(c)** YAP1 expression in the same patient cohort examined in Fig. 1a. Scale bar, 50 μm. Left panel, representative IHC image. Middle panel, comparison of YAP1 expression in matched tumour and non-tumour tissue. Right panel, YAP1 expression in tumours correlates positively with tumour grade and the ratio of RAF1/YAP1 expression in the same tumour negatively correlates with histological grade. **(d)** STAT3 expression in the same cohort. Left panel, representative IHC image. Middle panel, comparison of STAT3 expression in matched tumour and non-tumour tissue. Right panel, RAF1/YAP1 expression in the same tumour negatively correlated with the presence of medium-large clusters of STAT3 nuclear staining. Scale bar 50 μm. In (**a,b**), the data are represented as mean ± s.e.m., *$P \leq 0.05$, **$P < 0.01$, ***$P < 0.005$ according to Student's *t*-test. (**c,d**) Middle panels, In the box and whiskers plots (Tukey method), the box represents interquartile range, the middle bar the median, and the whiskers extend to 1.5 times the interquartile range. Statistical analysis was done using Wilcoxon signed rank test; the analysis in the right panels represents the Spearman correlation. $r_s$ and $P$ values are indicated. See also Supplementary Fig. 4.

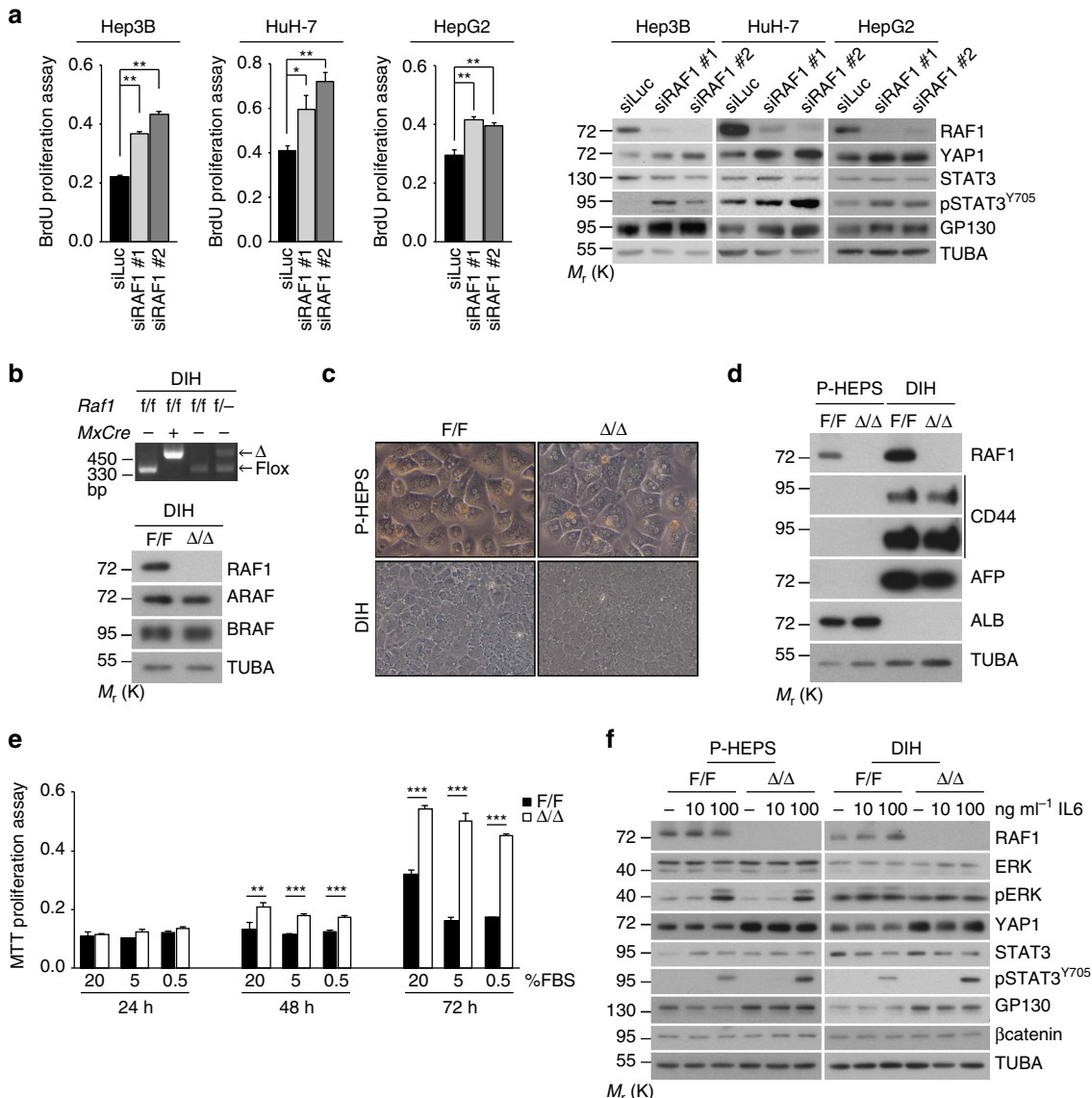

**Figure 4 | Molecular characterization of RAF1-deficient cells. (a)** siRNA-mediated RAF1 silencing promotes the proliferation of Hep3B ($n = 6$), HuH-7 ($n = 5$) and HepG2 ($n = 4$) cells and increases the expression of YAP1 and GP130 as well as STAT3 phosphorylation. siRAF1#1 targets the region around nucleotide 721, while siRAF1#2 is a mixture of siRNAs targeting the region from nucleotide 692 to 1,093 in the RAF1 mRNA. **(b)** PCR and immunoblotting analysis of F/F and RAF1Δ/Δ DIH. **(c)** Morphology (x200 magnification) and molecular characterization **(d)** of primary hepatocytes (P-HEPS) compared to DIH (AFP, α-fetoprotein; ALB, albumin; TUBA, loading control). The immunoblot is representative of two independent experiments. **(e)** Proliferation of DIH in decreasing amounts of FBS. **(f)** Molecular defects of RAF1-deficient P-HEPS and DIH treated with the indicated concentrations of IL6 for 30 min. TUBA serves as loading control. Data are presented as mean ± s.e.m. *$P \leq 0.05$, **$P < 0.01$, ***$P < 0.005$ according to Student's $t$ test. See also Supplementary Fig. 5.

Taken together, the results show that the molecular defects observed in RAF1-deficient tumour-bearing livers and in human HCC lines were already present in P-HEPS and did not arise during transformation or immortalization.

**YAP1 and STAT3 drive proliferation in RAF1 knockout cells.** In siRAF1 Hep3B cells, both YAP1 and pSTAT3 were found in the nucleus; accordingly, the target genes *CTGF* (YAP1 target) and *BIRC5* (common YAP1 and STAT3 target) were expressed at higher levels in these cells (Fig. 5a,b). Reducing YAP1 levels by siRNA (Fig. 5b) or, respectively, treating cells with the potent JAK kinase inhibitor Pyridone 6 (ref. 26) (P6; Fig. 5c), reduced both the expression of the target genes and cell proliferation. Importantly, P6 abrogated STAT3 phosphorylation but did not have any effects on YAP1 expression or ERK phosphorylation (Fig. 5c).

siRNA-mediated YAP1 silencing and P6 treatment also efficiently impaired target gene expression and proliferation of Δ/Δ DIH cultured in medium with 5% fetal bovine serum (FBS) (Fig. 5d and e), under which conditions ERK phosphorylation was clearly lower in Δ/Δ than in control DIH. Thus, the YAP1/STAT3 activation observed in siRAF1 Hep3B and Δ/Δ DIH contributes to proliferation. In intestinal epithelia, GP130 participates in IL6-driven STAT3 phosphorylation[27] as well as in YAP1 activation through phosphorylation of Y357 by Src (ref. 28). Knockdown of GP130 in premalignant hepatocytes and Hep3B cells reduced STAT3 phosphorylation but did not alter YAP1 expression or phosphorylation, indicating that GP130 is not required for YAP1 activation in these cells (Fig. 5f).

The results in Fig. 5d and e indicated that reduced ERK phosphorylation in Δ/Δ DIH cells grown in 5% FBS medium did not impair proliferation. To more directly assess the relevance of

ERK activation in the proliferation of DIH, we treated F/F and Δ/Δ DIH with GDC-0879, a potent and specific RAF inhibitor, or with the multikinase inhibitor Sorafenib, used in the treatment of advanced HCC. GDC-0879 completely inhibited ERK activation in DIH of either genotype, but affected the proliferation of the control much more than that of the Δ/Δ cells (Supplementary Fig. 7). In contrast, Sorafenib at the concentration used activated rather than inhibited ERK, but effectively reduced STAT3 phosphorylation as previously described[29] as well as DIH proliferation, independently of the genotype (Supplementary Fig. 7).

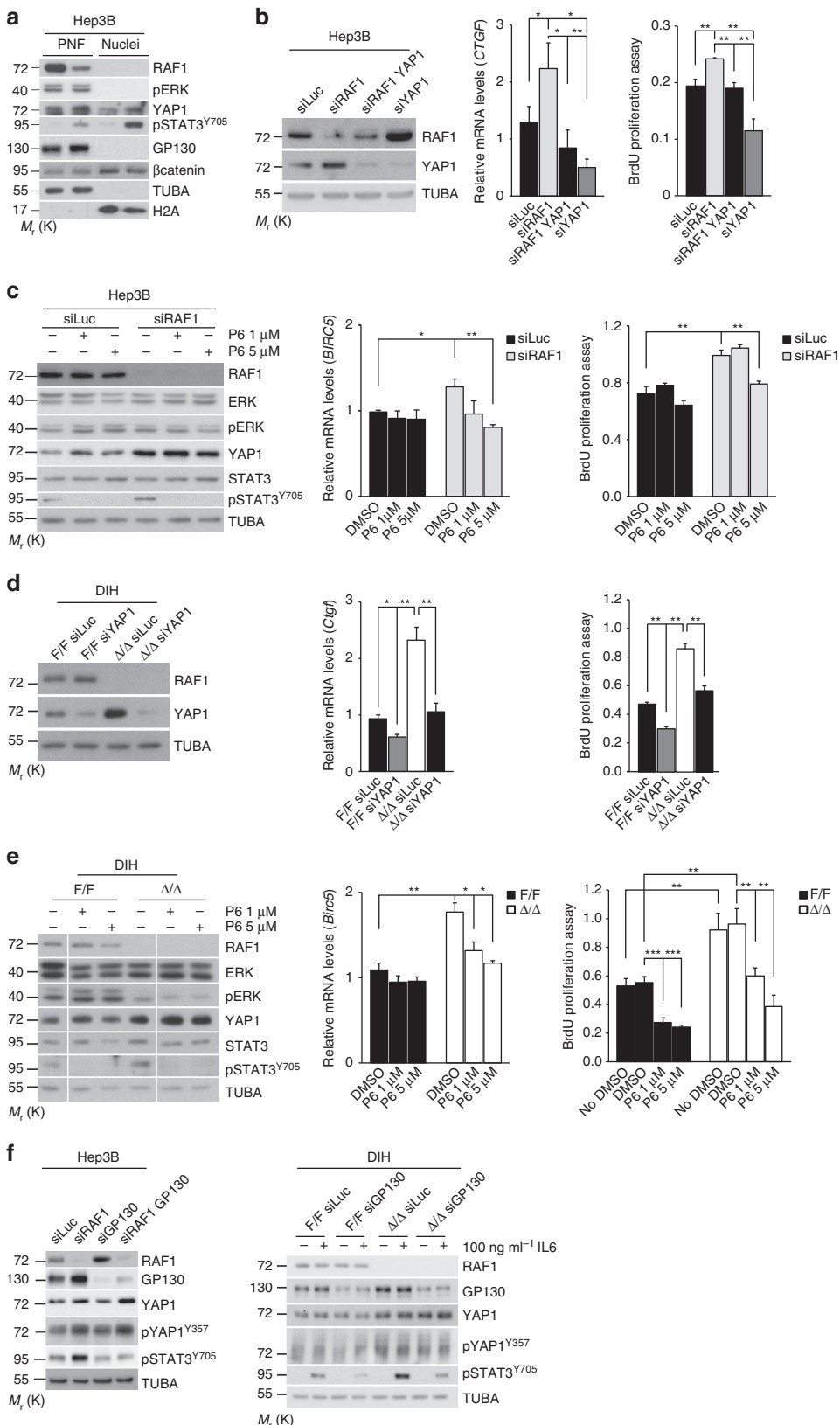

Thus, ERK phosphorylation status did not correlate with proliferation in DIH. We next treated DIH cells with PP2 to determine whether SFK inhibition had an impact on YAP1 phosphorylation on Y357, as recently described for intestinal cells[28], and whether this correlated with reduced proliferation. PP2 had a profound effect on cell proliferation, abolished the phosphorylation of YAP1 in DIH of both genotypes, and reduced STAT3 phosphorylation in RAF1 knockout DIH (Supplementary Fig. 7). Collectively, the data indicate that the inhibitors affect their expected targets; in addition, the differences observed in the inhibitors' effects in RAF1-proficient or -deficient DIH suggest a rewiring of signalling pathway in the RAF1 knockout DIH.

Our data so far are consistent with a model in which the regulation of YAP1 and STAT3 signalling underlies increased proliferation of RAF1-deficient Hep3B and DIH. This correlates with increased expression of YAP1 itself and of the STAT3 activator GP130. This increased expression did not correlate with higher mRNA amounts in Hep3B, P-HEPS, or DIH (Fig. 6a–c), suggesting a regulation at the translational or post-translational level. Blocking protein translation with cycloheximide revealed that YAP1 and even more so GP130 were subject to rapid turnover, and that both proteins were more stable in all three RAF1-deficient cell types (Fig. 6d–f).

## Discussion

Our study defines a tumour suppressor role of RAF1 in hepatic carcinogenesis. Firstly, RAF1 is found downregulated in a non-selected cohort of human HCC samples; secondly, modelling RAF1 downregulation in human HCC cells, in culture or in xenografts, increases cell proliferation; and thirdly, the same results are obtained in two independent genetic models (AlfpCre and MxCre-induced RAF1 ablation in the DEN/PB-treated mice) and in premalignant hepatocytes derived from these models. This consistency is remarkable given the molecular heterogeneity of human HCC as well as of the human cell lines studied[30] and the transgenic models used. The finding was entirely unexpected as the existing literature unanimously points to pro-tumourigenic functions of RAF1. RAF1 antagonizes apoptosis in both embryonic[9] and adult liver[10], and is required to promote proliferation in RAS-driven skin and lung carcinogenesis[31–33].

The molecular correlate of RAF1 ablation/downregulation is also remarkably consistent: lack of RAF1 results in the increased expression of YAP1 and GP130 and in STAT3 phosphorylation/ activation in all models tested. These include P-HEPS treated with IL6, showing that the defect is directly related to RAF1 deletion and does not arise during the transformation of RAF1-deficient cells. The phenotype is in line with the role of the IL6 pathway in the development and progression of cancer progenitor cells[7] and of YAP1 as antagonist of hepatocyte differentiation[20].

In contrast to models of transgenic YAP1 expression or of YAP1 activation by Hippo pathway disruption[3], RAF1 ablation does not lead to spontaneous hepatocarcinogenesis. This milder phenotype may be due to the Hippo pathway, which is functional in RAF1-deficient livers and cells and can counteract hepatocyte proliferation driven by mitogens[34]. In the autochthonous tumour models, the severity of the phenotype also correlates with the extent of liver inflammation, being stronger in Δhep than in Δp/np animals. This is consistent with the fact that increased YAP1 expression promotes hepatocyte proliferation in vivo only upon liver injury or inflammation, that IL6 cooperates with YAP1 in this setting[35], and that GP130 is necessary for full-fledged DEN-induced tumourigenesis in the mouse[14].

The ERK activation status, on the other hand, did not correlate with proliferation in autochthonous tumours, xenografts, or cultured cells, implying that in the absence of RAF1 proliferative signalling is rewired to rely on the activation of YAP1 and STAT3 rather than ERK. While ERK activation is widely regarded as pro-tumourigenic, it was recently shown to inversely correlate with stem cell self-renewal in mammary tumours[36] and with the maintenance of stem cell identity in mouse intestine[37]. Also noteworthy in this context, activation of the STAT3 pathway by IL6 (ref. 38) or EGFR/SFK (ref. 39) can render BRAF mutant cancer cells resistant to RAF/ERK inhibition.

Mechanistically, our data are consistent with a model in which RAF1 ablation promotes the expression of YAP1 and GP130, which in turn supports the activation of STAT3 by JAK, engendering a positive feedback loop supported by the inflammatory environment in which hepatocarcinogenesis occurs. Increased GP130 expression selectively supports the activation of STAT3 by proinflammatory cytokines of the IL6 family, but not by those of the IL10 family, broadly speaking anti-inflammatory in nature; the importance of the GP130/STAT3 axis in epithelial inflammation and gastrointestinal tumourigenesis[22] and of GP130 in liver tumours[4,7,14,40,41] has been amply documented. Besides the cell-autonomous proliferation phenotype, RAF1-deficient DIH are much more efficient than controls in attracting macrophages; they also produce higher amounts of CCL2, a STAT3 target gene[42], and of CXCL1, upregulated by YAP1 in breast cancer cell lines[43]. This is consistent with the increased numbers of inflammatory cells and the rich chemokine/cytokine milieu observed in Δhep tumour-bearing livers, and is reminiscent of the inflammation and macrophage accumulation caused by liver-restricted Hippo pathway inactivation[44,45]. In Δp/np animals, the failure of RAF1-deficient macrophages to migrate in response to chemokines and infiltrate the tumour-bearing livers correlates with limited inflammation and tumour load. This is consistent with the tumour-promoting role of inflammation in the DEN/Pb model, and implicates the immigrant macrophages as source of

**Figure 5 | Effect of YAP1 silencing, the P6 JAK inhibitor and GP130 silencing on DIH and Hep3B proliferation.** (**a**) siRNA-mediated RAF1 silencing in Hep3B cells increases YAP1 and GP130 expression and STAT3 activation without impacting ERK phosphorylation or β-catenin expression/localization. Immunoblot analysis of post-nuclear fraction (PNF; 20 μg, about 8% of total) and nuclear fraction (Nuclei; 20 μg, about 15% of total). (**b**) Silencing of YAP1 in RAF1-proficient and -deficient Hep3B cells (left panel, representative immunoblot analysis) downregulates the expression of the YAP1 target gene CTGF (middle panel, qPCR analysis) and reduces proliferation (right panel). (**c**) Treatment with the JAK inhibitor P6 abrogates STAT3 phosphorylation without impacting ERK phosphorylation or YAP1 expression (left panel, representative immunoblot analysis), decreases BIRC5 expression (middle panel, qPCR analysis) and reduces proliferation in RAF1-deficient Hep3B cells (right panel). (**d,e**) Similar results are obtained by subjecting RAF1-proficient and -deficient DIH to YAP1 silencing (**d**) or P6 treatment (**e**). (**f**) GP130 silencing decreases STAT3 phosphorylation but does not affect YAP1 expression or phosphorylation. Proliferation was assessed 48 h after siRNA transfection (with the exception of **c**, in which P6 was added 24 h after transfection and proliferation was measured after additional 48 h), gene expression after 24 h, and for immunoblotting cells were lysed after 1 h inhibitor treatment. In (**f**) DIH were treated for 30 min with the indicated concentration of IL6. Experiments were performed in DMEM supplemented with 10% FBS (Hep3B cells) or in DIH medium supplemented with 5% FBS (DIH). The immunoblots are representative of two independent experiments; TUBA was used as loading control. The plots represent the mean ± s.e.m. of three independent experiments. *P ≤ 0.05, **P < 0.01 according to Student's t test.

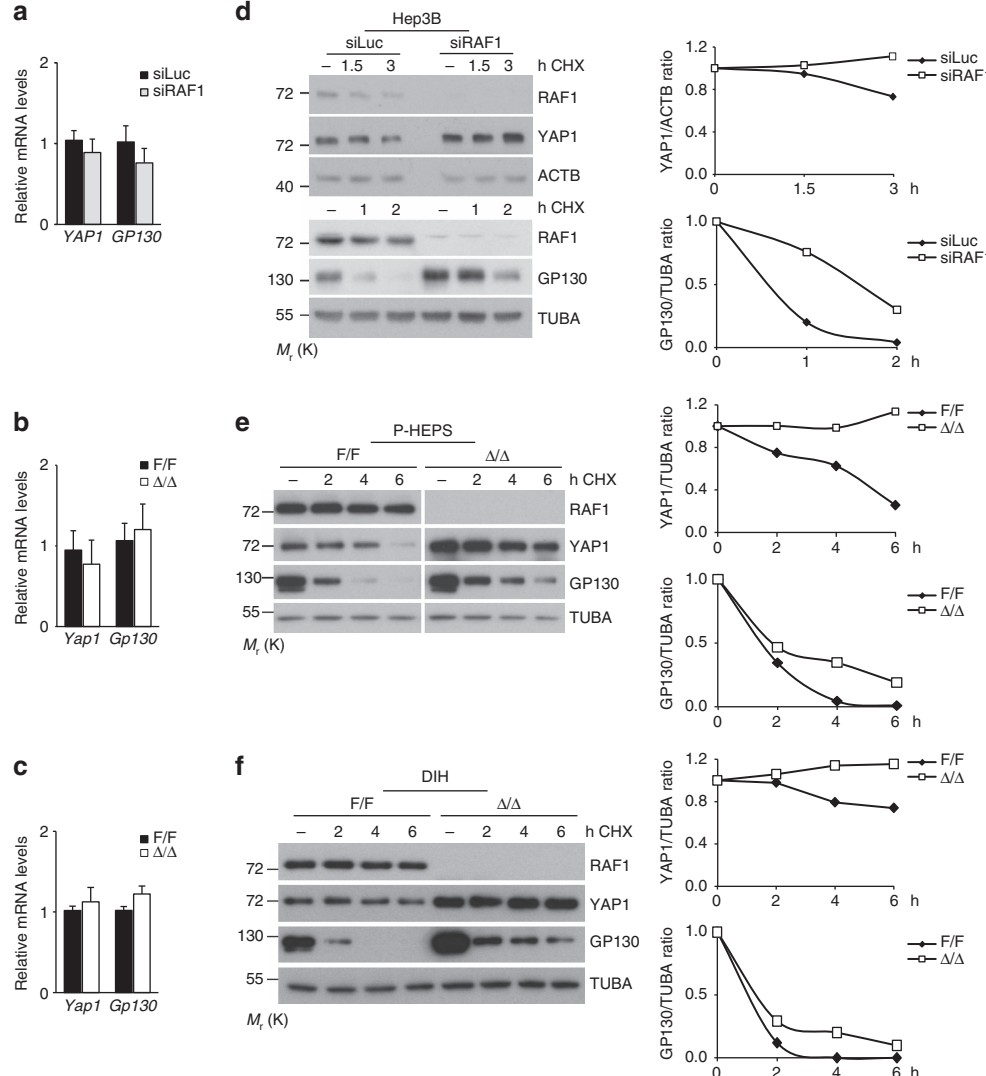

**Figure 6 | RAF1 ablation correlates with decreased YAP1 and GP130 protein turnover in Hep3B cells, primary hepatocytes (P-HEPS), and DIH.**
(**a–c**) qPCR analysis showing the expression of the YAP1 and Gp130 genes in Hep3B (**a**), P-HEPS (**b**) and DIH (**c**). qPCR data represent the mean (± s.e.m.) of three independent experiments; according to Student's *t* test. (**d–f**) Cells were treated with cycloheximide for the indicated amount of time prior to lysis. YAP1 and GP130 expression levels were determined by immunoblotting. A quantification is shown in the right panel; the amount of protein present in each of the untreated samples (normalized to TUBA or ACTB as loading controls) is set as 1.

inflammatory mediators. The competitive advantage of RAF1-deficient initiated hepatocytes is still evident in the Δp/np animals in the form of increased tumour multiplicity.

One key question is how RAF1 ablation increases YAP1 and GP130 expression at the protein level. YAP1 turnover is regulated by at least two ubiquitylation-dependent mechanisms: degradation by the βTrCP-SCF ubiquitin ligase complex, promoted by the Hippo pathway[46]; or by Elongin B/C/Cullin-5, antagonized by oncogenic RAS[47]. RAF1 is a Hippo pathway antagonist and a RAS effector; therefore, if its ablation impinged on one of these mechanisms, it should decrease, not increase YAP1 expression. YAP1/TAZ are also integral components of the βcatenin destruction complex, to which they recruit βTrCP, promoting βcatenin degradation in the absence of Wnt signals[48]. Considering that βcatenin expression and nuclear localization are not altered by RAF1 ablation (Fig. 3a, Supplementary Fig. 4a, Fig. 5a), it is unlikely that RAF1 regulates this complex; alternatively, RAF1 may regulate YAP1 turnover by ubiquitin-independent mechanisms, such as autophagy[49]. GP130 degradation is less well investigated. In the absence of IL6, basal turnover of GP130 is mainly maintained by the proteasome system, whereas after IL6 stimulation GP130 is monoubiquitinated by the E3 ligase c-Cbl and undergoes internalization, endosomal sorting and lysosomal degradation[50]. RAF1 can be recruited to endosomal membranes[51] and may therefore affect GP130 turnover by controlling either GP130 internalization or endosome trafficking.

Irrespectively of the precise mechanism, our data consistently show that reduced RAF1 expression confers on liver cells the double selective advantage of higher STAT3 activation (by GP130-dependent stimuli) and higher YAP1 expression. This two-birds-with-one-stone mechanism can apparently cooperate with a range of oncogenic mutations.

## Methods
**Animal studies.** *MxCre;Raf1^{F/F}* mice have been previously described[15] and were used to generate mice with a global RAF1 deletion (Δp/np mice; RAF1 deletion *in vivo* was induced by Poly I:C treatment). Mice lacking RAF1 in the hepatocyte compartment (Δhep mice), were generated by mating *Raf1^{F/F}* to *AlfpCre* transgenic mice[52]. All strains were on a Sv/129 background. To induce carcinogenesis, male

mice were injected with DEN (Sigma; 100 mg kg$^{-1}$ body weight i.p.) at 4 weeks of age, and received a Pb diet to promote tumour growth (Ssniff; 0.07% Pb, Sigma) from 8 weeks of age until killed. Animal experiments were authorized by the Austrian Ministry of Science, Research and Economy.

Hep3B xenografts ($1 \times 10^7$ in 0.1 ml of PBS) were inoculated in the flank of nude mice. shLuc or shRAF1 expression was induced by adding doxycycline to the drinking water (1.5 mg ml$^{-1}$ in dark bottles renewed every other day). Tumours were collected 40 days after injection and their volumes were determined according to the formula $V$ (in mm$^3$) $= a \times b^2/2$, where $a$ is largest diameter and $b$ is the perpendicular diameter.

**Human HCC samples.** 3 µm-thick sections of formaldehyde-fixed and paraffin-embedded samples of a non-selected cohort of surgically resected HCCs were obtained from the Biobank of the Medical University of Graz. All samples that fulfilled basic quality criteria (tumour cell content and absence of necrosis) were included in the study and in the data analysis. Collection and use of the samples was approved by the Ethical Committee of the Medical University of Graz (approval no. 27-334 ex14/15).

**Immunohistochemistry and immunoblotting.** H&E, TUNEL staining, immunohistochemistry and immunoblotting were carried out as described[33]. The following antibodies were used for immunohistochemistry: α-mouse, CD44 (550538, BD Biosciences, 1:50); Ki67 (Novocastra, 1:1,000), YAP1 (4912, Cell Signaling, 1:200), F4/80 (MCA497G, AbD Serotec, 1:50), FSP1 (27957, Abcam, 1:300), CD3 (A0452, DAKO, 1:1,000), βcatenin (32572, Abcam,1:500); α-human YAP1 (12395, Cell Signaling, 1:1,000), RAF1 (154754, Abcam, 1:1,500) and STAT3 (9139, Cell Signaling, 1:800). Granulocytes were visualized using Naphthol AS-D Chloroacetate (specific esterase) kit (91C-1KT, Sigma). Images were acquired with a ZEISS microscope Imager M1 ($\times$ 20/0.5 or $\times$ 10/0.3 Plan-NeoFluar objectives) equipped with ZEISS AxioCamMRc5 and ZEISS Axiovision Release 4.8.1 software. RAF1 and YAP1 expression in human samples was quantified by measuring reciprocal chromogen intensity with the ImageJ software as previously described[53].

For immunoblotting, P-HEPS, DIH and organs were lysed in RIPA buffer (150 mM NaCl, 5 mM EDTA pH 8, 1 mM EGTA pH 8, 0.1% Triton X-100, 50 mM Tris pH 8, sodium deoxycholate 0.5%, 1 µg ml$^{-1}$ pepstatin, 1 mM β-glycerophosphate). Hep3B, HuH-7 and HepG2 cells were lysed in 140 mM KCl, 3 mM MgCl$_2$, 1% NP-40, 0.2% SDS, 20 mM HEPES pH 7.4, 1 mM EDTA, 1.5 mM EGTA. Both buffers were supplemented with 1 mM phenylmethyl sulphonyl fluoride, 10 mM NaF, 1 mM Na$_3$VO$_4$ and Roche protease inhibitors cocktail. To obtain nuclear extracts, Hep3B were lysed in nucleus buffer (1 mM K$_2$HPO$_4$, pH 6.4, 150 mM NaCl, 14 mM MgCl$_2$, 1 mM EGTA, 0.1 mM DTT, 0.1% Triton X-100) and centrifuged at 450$g$ (10 min, 4 °C). Pellets were washed three times with nucleus buffer and lysed in lysis buffer.

The following antibodies were used for immunoblotting (1:1,000 unless otherwise stated): YAP1 (4912, 1:500), pYAP1-S127 (4911), pLATS1-T1079 (8654), LATS1 (3477), pMST1(T183)/MST2(T180) (3681), MST2 (3952), pERK1/2 (9101), ERK1/2 (9102), pSTAT3$^{Y705}$ (9145), STAT3 (9139), PCNA (2586) all Cell Signaling Technology; GP130 (sc-656), ACTB (sc-1616), pCFL1$^{S3}$ (sc-12912), ALB (sc-50536), ARAF (sc-408) BRAF (sc-9002), RAF1 (sc-133; all Santa Cruz Biotechnology; RAF1 (610152, 1:500), CD44 (550538), βcatenin (610153) from BD Biosciences; AFP (46799) and pYAP1$^{Y357}$ (62751) from Abcam; ROKα (04-841, Millipore)and TUBA (T9206, Sigma, 1:10,000). Immunoblots (representative of at least two experiments) were quantified using the ImageJ software. Uncropped blots are presented in Supplementary Fig. 8.

**Cell culture and cell-based assays.** P-HEPS were isolated from perfused livers (Liver perfusion medium and Liver digest medium, Gibco) and plated on collagen-coated dishes. P-HEPS were isolated from 8 to 12 weeks old F/F and Δp/np mice and cultured in DMEM supplemented with 5% FBS, 1% penicillin–streptomycin, 30 ng ml$^{-1}$ TGFα, 20 ng ml$^{-1}$ IGF-I and 0.7 nM insulin (all from Sigma). To obtain DIH cell lines, P-HEPS were isolated from $MxCre;Raf1^{F/F}$ or $Raf1^{F/F}$ mice 4 months after DEN injection and cultured for 3–4 weeks in DIH medium (DMEM containing 20% FBS, 0.01 g l$^{-1}$ insulin, 0.01 g l$^{-1}$ hydrocortisone hemisuccinate (Sigma), 1% penicillin–streptomycin, 1% L-glutamine (Gibco), 1 mM Pb (4920, Caesar & Loretz GmbH) and 20 ng ml$^{-1}$ EGF (2028-EG, R&D Systems). At this stage, transformed hepatocytes started proliferating and fibroblasts were removed from the culture by differential adhesion. After immortalization, all cells were treated with 1,500 U ml$^{-1}$ IFNβ (12,400-1, PBL interferon source), which led to RAF1 deletion in vitro in the $MxCre;Raf1^{F/F}$ cells (termed DIH Δ/Δ). Unless otherwise stated, DIH cells were cultured in 20% FBS DIH medium.

Bone marrow-derived macrophages (BMDM) were obtained by isolating bone marrow cells from 8 to 12 weeks old $RAF1^{F/F}$ and Δp/np mice and culturing them in DMEM supplemented with 10% FBS, 15% L-conditioned medium, 1% penicillin–streptomycin for 10 days[15]. Primary mouse keratinocytes, epidermal lysates and HaCat cells[54], immortalized endothelial cells[55] and mouse embryonic fibroblasts (MEFs)[9] were obtained and cultured as described. Human HCC lines were obtained from Wolfgang Mikulits (Institute of Cancer Research, Medical

University of Vienna—verified by STR analysis) and cultured in DMEM or RPMI (HepG2) supplemented with 10% FBS and 1% penicillin–streptomycin. All cells lines were tested Mycoplasma negative (Myco Alert, Cambrex) within 6 months of performing the experiment.

Lipofectamine RNAiMax (Invitrogen) was used to transfect DIH or human HCC lines with RAF1 and YAP1 siRNAs. The siRNAs (all from SIGMA) were as follows: mouse RAF1: 40 nM esiRAF1, EMU036131; mouse YAP1, 40 nM esiYAP1, EMU088231; mouse IL6ST (GP130), 40 nM esiIL6ST, EMU005111; human RAF1: 30 nM, NM_002880 (ID: SASI_Hs01_00174876; RAF1#1) or 30 nM esiRAF1 EHU050131 (RAF1#2); human YAP1, 30 nM esiYAP1, EHU113021; human IL6ST (GP130), 30 nM esiGP130, EHU117331.

esiRNA against Renilla Luciferase (RLuc, EHURLUC) was used as negative control.

Unless otherwise stated, cells were assayed 48 h after transfection.

Isogenic cell lines stably expressing doxycycline-inducible RAF1 shRNA (5′TCGAGGTGTGCGAAATGG AATGAGCTTCAAGAGAGCTCATTCCATTT CGCACACTTTTTTACGCGTA3′) or shRNA targeting Luciferase (cloned in the Tet-on inducible expression vector, Clontech) were generated by transfection with Turbofectin 8.0 (Origene) followed by antibiotic selection. Silencing was induced by adding 2 µg ml$^{-1}$ of doxycycline to the medium.

Cell proliferation, determined by absorbance quantitation, was assessed by MTT (M5655, Sigma, 570 nm) or by BrdU proliferation assay (2750, Millipore, 450 nm). $5 \times 10^3$ cells per well were plated in 96-well plates in 5% (DIH) or 10% FBS medium (human HCC lines). In selected experiments, cells were treated with the chemical inhibitors P6 (Calbiochem), GDC-0879 (Selleckchem), Sorafenib (Selleckchem), PP2 (Sigma) or cycloheximide (Abcam; 25 µg ml$^{-1}$ for Hep3B cells, 100 µg ml$^{-1}$ for P-HEPS and DIH).

Stimulation with 10 or 100 ng ml$^{-1}$ IL6 (ProSpec) was carried out in 5% FBS (P-HEPS) and 20% FBS (DIH) for 30 min.

**Flow cytometry of liver fractions.** P-HEPS were isolated from F/F and Δp/np mice 4 months after DEN injection. Non-aggregate and aggregate fractions were separated based on the ability of the cell suspensions to pass through a 40 µm cell strainer. Each fraction was mechanically dispersed and cells were stained with Fixable Viability Dye eFluor 520 (65-0867, 1:1,500) and α-mouse CD44 (17-0441-81), CD45 (12-0451-81), Ter119 (12-5921-81) and CD31 (12-0311-81), all diluted 1:100. All reagents were from Affymetrix eBioscience. Samples were measured on FACSCalibur and analysed with FlowJo.V10.

**FlowCytomix analyte assay.** Chemo- and cytokines in cell supernatants, serum samples and liver tissue lysates were detected using the Affymetrix eBioscience bead-based multiplex immunoassay. Data were analysed with FlowCytomix Pro2.4 software.

**Quantitative PCR.** RNA from P-HEPS, DIH and Hep3B cells was isolated using Nucleospin RNA II kit (Macherey-Nagel). cDNA was prepared using Oligo(dT)$_{18}$ primer, dNTPs, and RevertAidReverse Transcriptase (Thermo Scientific). qPCR was performed using Go Taq qPCR Master mix (Promega). Relative expression was calculated by the ΔΔCT method using $ACTB$ as housekeeping gene. The primers used are: mouse primers: $Yap1$ forward (5′GTCCTCCTTTGAGATCCCTGA3′); reverse (5′TGTTGTTGTCTGATCGTTGTGAT3′); $gp130$ forward (5′CTTTGGGCAGATCGGAGCAGAA3′); reverse (5′CCCTCATTCACAATGCAAGTCA3′); $Ctgf$ forward (5′AGAACTGTGTACGGAGCGTG3′); reverse (5′GTGCACCATCTTTGGCAGTG3′), $Birc5$ forward (5′AGAACAAAATTGCAAAGGAGACCA3′); reverse (5′GGCATGTCACTCAGGTCCAA3′); $ActB$ forward (5′CCTCTATGCCAACACAGTGC3′); reverse (5′GTACTCCTGCTTGCTGATCC3′). human primers: $YAP1$ forward (5′CCCGACAGGCCAGTACTGAT3′); reverse (5′CAGAGAAGCTGGAGAGGAATGAG3′): $GP130$ forward (5′GACCATCTAAAGCACCAAGTTTCT3′); reverse (5′AAAGGAGGCAATGTCTTCCACA3′); $CTGF$ forward (5′CCTTCCCGAGGAGGGTCAA3′); reverse (5′CAGTCGGTAAGCCGCGAG3′); $BIRC5$ forward (5′ CTTTCTCAAGGACCACCGCA3′); reverse (5′ CTCGGC CATCCGCTCC3′), $ACTB$ forward (5′AGAGCTACGAGCTGCCTGAC3′); reverse (5′AGCACTGTGTTGGCGTACAG3′). All primers were from Sigma.

**Migration assays.** BMDM ($1 \times 10^5$ per well, triplicates) were allowed to migrate towards CCL2 (10 ng ml$^{-1}$, 479JE, R&D Systems) or DIH ($7 \times 10^5$ per well in 0.5% FBS DIH medium) through a transwell membrane (pore size 8 µm, BD Falcon) in DMEM containing 0.5% FBS. Where indicated, cells were pre-treated with ROK inhibitor (Y27632, Calbiochem, 10 µM, 30 min). Six hours after plating, the cells were fixed (4% paraformaldehyde (PFA)), stained with crystal violet and counted. For in vivo migration assays, 30 ng CCL2 in 300 µl matrigel plugs (356231, BD Biosciences) were injected into the flanks of 8–12 weeks old F/F and Δp/np mice, removed and analysed 5 days later.

**Statistical analysis.** Animal experiments were performed comparing littermates, the evaluators were aware of animal identity throughout the experiments and outcome assessment. The log-rank test was used to evaluate the significance of the difference in survival (Supplementary Fig. 1c). Where applicable, power calculations were used to determine the sample size necessary to obtain significant results ($P < 0.05$) with a power of $> 0.80$, assuming twofold changes and a s.d. of 10%. Histological samples were analysed by counting or measuring at least five microscopic fields/section. The investigator was blinded to group allocation. For experiments involving cultured cells, unless otherwise stated values are expressed as means ± s.e.m. of three independent experiments; $P$ values were calculated with the two-tailed Student's $t$-test, hetero- or homoskedastic as determined by a previous $F$-test of equality of variances. The human immunohistochemical data was analysed using Wilcoxon signed rank test and Spearman correlation. A $P$ value $\leq 0.05$ is considered statistically significant.

**Data availability.** The authors declare that all data supporting the findings of this study are available within the paper and its Supplementary information files.

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

## Acknowledgements

We thank Vitaly Sedlyarov, Josip Jeric and Martina Mittlboeck for help with the statistical analysis, Clemens Bogner, Karin Ehrenreiter, Iris Kufferath, Daniela Pabst and the animal facility for excellent technical assistance. The work was supported by FWF grants SFB021 and W1220-B09, INFLA-CARE and GROWTHSTOP (European Commission) and by the Obermann-Stiftung, all to M.B.; by grant INDICAR (Mahlke-Obermann Stiftung & European Commission grant 609431) to ED; and by the Christian Doppler Laboratory for Biospecimen Research and Biobanking Technology.

## Author contributions

I.J., G.M., J.R., E.D. and A.L.C. designed, carried out and interpreted experiments. M.P. and B.T. helped with the acquisition of data. I.J. prepared the data for publication. I.F. helped with the histology, K.Z. contributed to the design and interpretation of the human studies, M.B. designed and supervised the project, helped with data interpretation, and wrote the manuscript.

## Additional information

**Competing financial interests:** The authors declare no competing financial interests.

