## [Peer Review File · Nature Communications]

Reviewers' comments:

Reviewer #1 (Remarks to the Author):

Using the DEN model of hepatocarcinogenesis in mice Jeric and colleagues describe an unexpected anti-proliferative role of RAF1 in HCC development. Downregulation of RAF by RNA interference or genetic loss of RAF1 in vivo promotes hepatocellular tumor cell growth. The authors propose that this pro-tumorigenic phenotype in the absence of RAF function is linked to an upregulation of gp130-dependent Stat3 activation and YAP1 upregulation. Pharmacological Stat3 inhibition and YAP1 knockdown revert the elevated proliferation in RAF1 deficient cells.

This is a very interesting and very well performed study that may have great impact for our understanding and the treatment of human disease.

I only have two minor points that should be considered.

1. Recently, it was suggested in intestinal epithelial cells that gp130 triggers YAP1 activation independently of STAT3 via Src and Yes. In the manuscript here only a pharmacological Stat3 inhibitor was used, however, it was not addressed whether increased YAP1 expression could be linked to gp130 as well. Thus, it would be nice if the authors could provide evidence using if siRNA mediated knockdown of gp130 in Raf1 deficient cells that YAP1 is downstream of gp130 or not.
2. To corroborate the potential importance for human diseases and sorafenib treatment in human patients, the manuscript would benefit if the authors could examine whether the phenotype proposed here - particularly gp130 and YAP1 upregulation- can be found in sorafenib treated patients or at least in the mouse models used in this study. Presumably it will be important to compare the effects of sorafenib in vitro and in vivo considering that concomitant loss of Raf1 in hepatocytes and hematopoietic cells shows a milder phenotype than deletion in hepatocytes only, suggesting that RAF1 inhibition in myeloid cells may actually be tumorsuppressive.

Reviewer #2 (Remarks to the Author):

The manuscript by Jeric et al. reveals a robust and unanticipated contribution of RAF1 to tissue homeostasis via support of HIPPO pathway activity. The phenotypes in mice are significant and convincing. The cell biology confirms cell autonomous effects. Overall, the manuscript represents an important body of work that should be shared with the community at large. I have a few suggestions:

1. Impact: can the investigators inform the audience more effectively with respect to the overarching relevance of the observations? I.e.- is the mechanistic relationship at the molecular level generalizable to all tissues/cell types, with selective consequences or is there tissue/cell type selective mechanistic coupling of Raf1 and YAP? Related to this, in the tumors the authors describe as RAF1 dependent (lung?) – I think there is evidence that YAP1 activation is a targeted therapy resistance mechanism. Does this suggest the relationships shown in this paper are independent of RAF1 kinase activity?
2. Mechanism: this work is a genetic tour-de-force, however, some straight-forward molecular relationships could be pursued with the material at hand. I.e. potential roles of RASSF1A and/or unanticipated participation of Raf1 in MST1/2 activation?
3. RNAi: Many shRNA and siRNA experiments appear to be performed with a single reagent. RAF1 is well covered by orthoganol approaches. YAP1 is not

Reviewer #1 (Remarks to the Author):

Reviewer is very appreciative of our work, which s/he predicts to have a great impact for our understanding of hepatocarcinogenesis.

The Reviewer has only two minor points:

1. “it would be nice if the authors could provide evidence using if siRNA mediated knockdown of gp130 in Raf1 deficient cells that YAP1 is downstream of gp130 or not”.

The GP130 KD was performed in DIH and Hep3B cells. Unfortunately, the only commercially available pY357 antibody performs rather poorly in our experiments; however, the results show that neither YAP1 expression nor its basal or IL6-induced phosphorylation on Y357 are affected by the knockdown of GP130; as a positive control, pSTAT3^{Y705} levels are decreased under these conditions (Figure 5f of the revised version; discussed on page12 of the revised manuscript). However, the Src inhibitor PP2 reduces YAP1 tyrosine phosphorylation in DIH (Supplementary Fig. 7 of the revised version). Together, the data suggest that Src kinases activate YAP1 in a GP130-independent manner in premalignant and malignant hepatocytes.

2. “the manuscript would benefit if the authors could examine whether the phenotype proposed here - particularly gp130 and YAP1 upregulation- can be found in sorafenib treated patients or at least in the mouse models used in this study. Presumably it will be important to compare the effects of sorafenib in vitro and in vivo considering that concomitant loss of Raf1 in hepatocytes and hematopoietic cells shows a milder phenotype than deletion in hepatocytes only, suggesting that RAF1 inhibition in myeloid cells may actually be tumor suppressive.

This second point is about the impact of Sorafenib therapy on the mechanism described here. The reviewer suggests in vivo and in vitro experiments; systemic Sorafenib administration would impact many aspects of tumorigenesis, particularly increased angiogenesis which is a major feature of HCC and is targeted by Sorafenib (Horwitz et al., 2014). Therefore the experiments in vivo, besides being extremely time-consuming and involving animal suffering, are not likely to yield conclusive or readily interpretable results; it is also impossible for us to obtain access to a patient cohort and analyze it within the time allotted for revision. As proof of principle, we have treated premalignant DIH with a range of inhibitors: a specific RAF inhibitor (GDC-0879) and with Sorafenib. The results show that neither complete inhibition of RAF kinase activity by GDC-0879 nor paradox ERK activation by Sorafenib increase YAP1 or GP130 expression. Both Sorafenib and GDC reduced the proliferation of DIH, but GDC was less efficient in RAF1-deficient cells. The data are shown in Supplementary Fig. 7.

We hope that the reviewer will bear with us on this point.

Reviewer #2 (Remarks to the Author):

This Reviewer is also appreciative of the manuscript's novelty and importance. S/he has a few suggestions:

1a. "is the mechanistic relationship at the molecular level generalizable to all tissues/cell types, with selective consequences or is there tissue/cell type selective mechanistic coupling of Raf1 and YAP?"

We have now investigated the correlation between RAF1-deficiency and YAP1 expression in a number of systems. No differences were detected in WT and KO primary mouse keratinocytes or mouse epidermal cell lysates and in the human keratinocyte cell line HaCat. These were investigated because of the impact of RAF1 on tumorigenesis in this tissue. In addition, we tested immortalized mouse embryonic fibroblasts and endothelial cells and primary bone marrow derived macrophages. A slight increase in YAP1 can be detected in MEFs. Thus, the mechanism described appears to be selectively active in hepatocytes. These data are included in the revised version as Supplementary Fig. 6.

1b. "in the tumors the authors describe as RAF1-dependent (lung?) – I think there is evidence that YAP1 activation is a targeted therapy resistance mechanism. Does this suggest the relationships shown in this paper are independent of RAF1 kinase activity?"

The reviewer is probably referring to the paper by Lin et al. (Nature Genetics 2015, 47:250–256), showing that "Increased YAP1 in tumors harboring BRAF V600E was a biomarker of worse initial response to RAF and MEK inhibition". The paper shows that pathway inhibition and YAP1 knockdown converge on inhibiting the expression of BCL-xL, thus increasing apoptosis. It does not, however, report any mechanistic links between ERK pathway activity and YAP1 expression. It is therefore impossible, from these data, to predict whether the increase in YAP1 observed in RAF1-deficient cells would be due to kinase-dependent or independent functions of RAF1. We are currently investigating the connection between RAF1 activity and YAP1 overexpression in a series of structure/function studies which is outside of the scope of this manuscript; however, we have attempted to address this point by treating DIH with RAF and multikinase inhibitors. The results show that neither complete inhibition of RAF kinase activity by GDC-0879 nor paradox ERK activation by Sorafenib increase YAP1 or GP130 expression (Supplementary Fig. 7).

2. "some straight-forward molecular relationships could be pursued with the material at hand. I.e. potential roles of RASSF1A and/or unanticipated participation of Raf1 in MST1/2 activation"?

We started this work based on the assumption that the RAF1 KO would negatively influence hepatocarcinogenesis and YAP1 expression because of RAF1's known ability to suppress MST2. In contrast, we found that the KO increased tumorigenesis and YAP1 expression, and therefore analyzed Hippo pathway components in particular detail. The data (Fig. 3a and Supplementary Fig. 4b) argue for a lack of interaction between RAF1 and the Hippo pathway in this system. We also have genetic data (Figure provided for the Reviewer 2's perusal) showing that RASSF1A does not play a role in DEN-induced liver carcinogenesis in 129/BL6 mice, in agreement with the previously reported finding that liver tumors were not observed in RASSF1A knockout mice of mixed background (Tommasi et al. 2005, Cancer research, 65(1):92-8). On the basis of these findings, we did not pursue the topic further.

3. RNAi: Many shRNA and siRNA experiments appear to be performed with a single reagent. RAF1 is well covered by orthogonal approaches. YAP1 is not

YAP1 silencing was achieved with esiRNA, which have a 10-fold higher target specificity than oligo siRNAs and are therefore considered the method of choice for gene silencing with minimal off-target effects (Kittler et al., 2007, Nat. Methods 4, 337-344,). In addition, the regions targeted by esiRNAs on human and mouse YAP1 are partially overlapping but not identical (human coding sequence: NT 389-1855, esiRNA target NT 1321-1805; mouse coding sequence NT 207-1673, esiRNA target NT 1529-2066); the fact that YAP1 silencing shows the same effects, both in terms of target gene expression and proliferation, in mouse DIH and human Hep3B increases the confidence in our results. Finally, although our manuscript itself does not contain orthogonal approaches, several previous studies have shown that YAP1 silencing or knockout reduces hepatocyte proliferation (Zhou et al. 2009, Cancer Cell 16:425-38; Fitamant et al. 2015, Cell Reports, 10:1692-1707; Su et al. 2015, eLife, 4:e02948), providing independent confirmation of our data. We thus hope that the reviewer will bear with us on this point.

REVIEWERS' COMMENTS:

Reviewer #1 (Remarks to the Author):

The authors have addressed my concerns adequately

Reviewer #2 (Remarks to the Author):

The response to review is somewhat disappointing. Mechanism remains sparse. The phenotype remains strong. The authors argue around most points. I remain at the original level of enthusiasm, however, the response to concerns about YAP validation was unacceptable in my opinion and exposes the authors to serious artifact liability. esiRNA is NOT a panacea for on-target validation by any means. YAP1 phenotypes must be validated by orthogonal methods. Ideally, this involves rescue by complementation (hard to do using esiRNA..). I imagine Nature has house standards with respect to this issue.

NCOMMS-16-14452A –point-by-point reply:

We thank both Reviewers for their careful assessment of the manuscript.

Below, we address the remaining point of Reviewer 2 (response in italics):

“The response to concerns about YAP validation was unacceptable in my opinion and exposes the authors to serious artifact liability. esiRNA is NOT a panacea for on-target validation by any means. YAP1 phenotypes must be validated by orthogonal methods. Ideally, this involves rescue by complementation (hard to do using esiRNA..). I imagine Nature has house standards with respect to this issue.”

It is unfortunate we couldn't convince Reviewer 2 on this point. In our case, Nature Communication has agreed to accept the use of esiRNA both in mouse and human cells as a proof that off target effect are excluded. We find this fully justified in view of the fact that several previous studies have shown that YAP1 silencing or knockout reduces hepatocyte proliferation (Zhou et al. 2009, Cancer Cell 16:425-38; Fitamant et al. 2015, Cell Reports, 10:1692-1707; Su et al. 2015, eLife, 4:e02948), providing independent confirmation of our data.